# MICM: Rethinking Unsupervised Pretraining for Enhanced Few-shot Learning

## ABSTRACT

Humans exhibit a remarkable ability to learn quickly from a limited number of labeled samples, a capability that starkly contrasts with that of current machine learning systems. Unsupervised Few-Shot Learning (U-FSL) seeks to bridge this divide by reducing reliance on annotated datasets during initial training phases. In this work, we first quantitatively assess the impacts of Masked Image Modeling (MIM) and Contrastive Learning (CL) on few-shot learning tasks. Our findings highlight the respective limitations of MIM and CL in terms of discriminative and generalization abilities, which contribute to their underperformance in U-FSL contexts. To address these trade-offs between generalization and discriminability in unsupervised pretraining, we introduce a novel paradigm named Masked Image Contrastive Modeling (MICM). MICM creatively combines the targeted object learning strength of CL with the generalized visual feature learning capability of MIM, significantly enhancing its efficacy in downstream few-shot learning inference. Extensive experimental analyses confirm the advantages of MICM, demonstrating significant improvements in both generalization and discrimination capabilities for few-shot learning. Our comprehensive quantitative evaluations further substantiate the superiority of MICM, showing that our two-stage U-FSL framework based on MICM markedly outperforms existing leading baselines. [1]

## CCS CONCEPTS

• **Computing methodologies** → **Computer vision**.

## KEYWORDS

Unsupervised Few-shot Learning, Contrastive Learning, Masked Image Modeling, Masked Image Contrastive Modeling

## 1 INTRODUCTION

Achieving high-level performance in deep representation learning typically requires large datasets, detailed labeling processes, and significant supervisory involvement. This requirement becomes even more daunting as the complexity of downstream tasks increases, challenging the scalability of supervised representation learning methods. In contrast, human learning is remarkably efficient, managing to acquire new skills from minimal examples

---

[1]Codebase is attached to the supplementary material and will be made publicly available upon acceptance.

*ACM MM, 2024, Melbourne, Australia*
© 2024 Copyright held by the owner/author(s). Publication rights licensed to ACM.
ACM ISBN 978-x-xxxx-xxxx-x/YY/MM
https://doi.org/10.1145/nnnnnnn.nnnnnnn

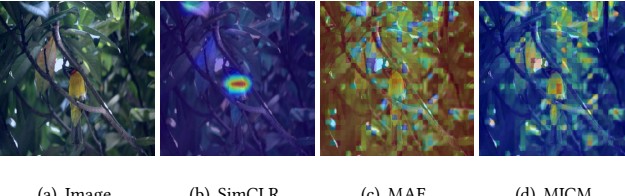

| (a) Image | (b) SimCLR | (c) MAE | (d) MICM |
|---|---|---|---|

**Figure 1: Illustrating the attention map with SimCLR [9] (CL), MAE [22] (MIM), MICM.**

with little supervision. Few-shot learning (FSL) [33, 40] aims to narrow this gap between human and machine learning capabilities. Although FSL has shown promising results in supervised settings, its dependence on extensive supervision remains a significant limitation. To address this, unsupervised FSL (U-FSL) [10, 32, 33] has been developed, mirroring the structure of supervised approaches. It involves pretraining on a wide dataset of base classes and then quickly adapting to novel, few-shot tasks. The primary goal of U-FSL pretraining is to develop a feature extractor that understands the global structure of unlabeled data, and subsequently tailors the encoder for new tasks. The increasing interest in U-FSL reflects its practicality and alignment with self-supervised learning methods, emphasizing its potential to significantly enhance machine learning processes.

Recent advancements in state-of-the-art (SOTA) methods have largely employed Contrastive Learning (CL) [10, 32, 33], particularly in transfer-learning scenarios. These methods have achieved impressive results across various benchmarks. As illustrated in Figure 2(a), the principle of contrastive representation learning [9] involves drawing 'positive' samples closer and pushing 'negative' ones away in the representation space. This technique focuses on specific objects within datasets, thus improving representational learning for image classification tasks, as depicted in Figure 1(b). Conversely, Masked Image Modeling (MIM) [12, 22] (Figure 2(b)) trains models to predict the original content of intentionally obscured image portions. This approach facilitates comprehensive learning of features across all image patches, including peripheral ones, as shown in Figure 1(c). In our research, we quantitatively assessed the impacts of MIM and CL on FSL tasks, revealing that *while CL prepares models to prioritize features typical in training datasets, potentially compromising reliability in novel categories*, and *MIM fosters a broad and generalized understanding of image features but struggles to develop discriminative features crucial for accurately categorizing new classes in few-shot scenarios*.

To explore the trade-offs between generalization and discriminability in unsupervised pretraining, we introduce Masked Image Contrastive Modeling (MICM), a novel method that ingeniously combines essential aspects of both CL and MIM to boost downstream inference performance. Illustrated in Figure 2(c), MICM utilizes an encoder-decoder architecture akin to MIM but integrates

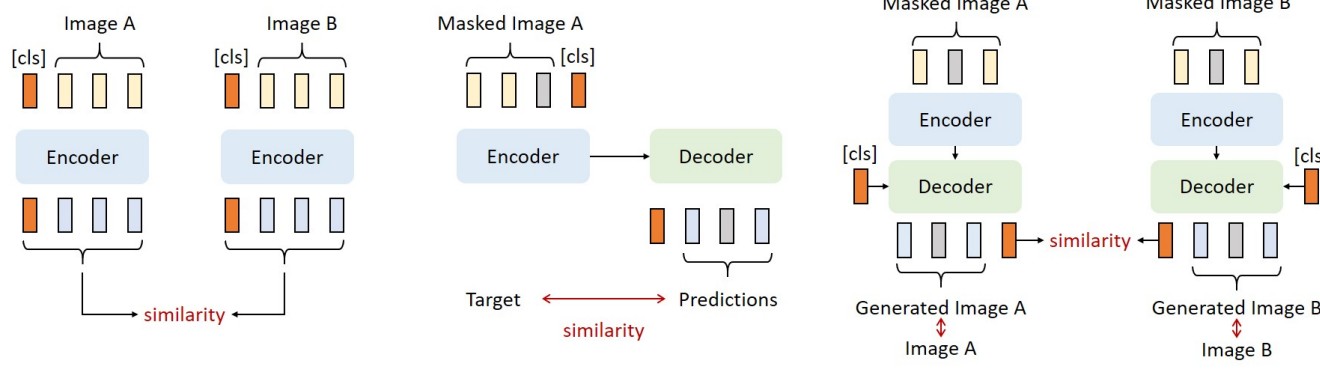

(a) Contrastive Learning  (b) Masked Image Modeling  (c) Masked Image Contrastive Modeling

Figure 2: (a) *Contrastive Learning* is dedicated to learning discriminative data representations by contrasting and differentiating between similar (positive) and dissimilar (negative) pairs of data samples. This approach emphasizes the relative comparison to achieve distinctiveness in the learned features. (b) *Masked Image Modeling* involves training a model to accurately predict the original content of intentionally obscured (masked) portions of images. This technique focuses on learning comprehensive and robust feature representations by encouraging the model to infer missing information. (c) *Masked Image Contrastive Modeling* implements a decoder that simultaneously enhances features and reconstructs the original content of images. This method synergistically merges the principles of contrastive representation learning with effective pretext task design, thereby integrating the strengths of both approaches to achieve more nuanced and effective learning.

a decoder that simultaneously enhances features and reconstructs images. This method not only merges contrastive learning with effective pretext task designs but also adapts efficiently to downstream task data. We further introduce a U-FSL framework with two phases: *Unsupervised Pretraining* and *Few-Shot Learning*. During *Unsupervised Pretraining*, MICM blends CL and MIM objectives in a hybrid training strategy. In the *Few-Shot Learning* phase, MICM adapts to various few-shot strategies, both inductive and transductive. Extensive experimental analysis confirms the benefits of MICM, showing it significantly enhances both the generalization and discrimination capabilities of pre-trained models, achieving top-tier performance on multiple U-FSL datasets.

To summarize, the main contributions of our paper are as follows:

- We reveal the limitations of MIM and CL in terms of discriminative and generalization abilities, respectively, to their consequent underperformance in U-FSL contexts.
- We propose Masked Image Contrastive Modeling (MICM), a novel structure that blends the targeted object learning prowess of CL with the generalized visual feature learning capability of MIM.
- Extensive quantitative and qualitative results show that our method achieves SOTA performance on several In-Domain and Cross-Domain FSL datasets.

## 2 RELATED WORK

### 2.1 Few-Shot Learning

Few-shot learning (FSL) in visual classification contends with the challenge of recognizing objects from very limited samples. Primarily, this task is approached through two principal methodologies: transfer learning and meta-learning. Transfer learning, as discussed by Tian et al. [40], leverages knowledge from models pre-trained

on large datasets to adapt to new, less-represented tasks. Meta-learning, alternatively, encompasses several strategies: model-based [6], metric-based [39], and optimization-based [1]. Model-based approaches focus on adapting the model parameters rapidly for new tasks. Metric-based methods compute distances between samples to facilitate class differentiation, while optimization-based strategies aim to maximize learning efficiency with scarce examples. Building upon inductive FSL, transductive FSL seeks to enhance real-world application by incorporating unlabeled data into the learning process for pre-classification tuning. Among the various techniques employed, graph-based methods such as protoLP [52] utilize graph structures to strengthen the information flow and relationships between support and query samples. Clustering-based approaches, exemplified by EASY [4], Transductive CNAPS [3], and BAVARDAGE [26], refine feature representations using advanced clustering techniques. Additionally, applications of the Optimal Transport Algorithm, like BECLR [33], are used to align feature distributions more effectively during testing phases. A novel approach, TRIDENT [38], integrates a unique variational inference network to enhance image representation in FSL scenarios.

### 2.2 Unsupervised Few-Shot Learning

U-FSL broadens the scope of unsupervised learning by requiring models to not only learn data representations without supervision but also to rapidly adapt to new few-shot tasks. This challenging dual requirement has led to the exploration of various innovative methods. Knowledge distillation and contrastive learning are notably effective in U-FSL, enhancing model adaptability through robust feature representations [27]. Furthermore, clustering-based approaches have demonstrated considerable success by optimizing data groupings to better model task-specific nuances [31]. Despite these advancements, many traditional unsupervised methods are

geared towards batch data processing, which may not seamlessly translate to the dynamic requirements of few-shot scenarios. To mitigate this, some strategies integrate meta-learning principles to generate synthetic training scenarios that improve data efficiency and model responsiveness [2]. However, such approaches can sometimes lead to suboptimal data usage [18]. Recent innovations in U-FSL also include the use of graph-based structures to map relationships within data [8], the application of Structural Causal Models (SCM) in the context-aware multi-variational autoencoder (CMVAE) [34], and the deployment of variational autoencoders (VAE) in frameworks like CMVAE and Meta-GMVAE [29]. Additionally, the exploration of rotation invariance in self-supervised learning enriches the robustness of models against geometric variations in few-shot learning tasks [45]. Another notable approach is MlSo, which leverages multi-level visual abstraction features combined with power-normalized second-order base learner streams to enhance the discriminative capability of models in U-FSL [49].

## 3 QUALITATIVE STUDY ON UNSUPERVISED PRETRAINING METHODS FOR U-FSL

This section delves into Unsupervised Few-Shot Learning (U-FSL), elucidating the task and assessing the impact of different unsupervised pretraining methodologies on model performance.

### 3.1 Unsupervised Few-shot Learning

U-FSL operates under a widely recognized protocol delineated in recent work [10, 27, 32, 33]. Initially, models undergo an unsupervised pretraining phase using a vast unlabeled dataset $D_{\text{base}} = \{x_i\}$, which encompasses a variety of *base* classes. Subsequently, the models are tested in a few-shot inference phase using a smaller, labeled test dataset $D_{\text{novel}} = \{(x_i, y_i)\}$ comprising *novel* classes, ensuring no overlap exists between the base and novel classes. Each few-shot task $\mathcal{T}_i$ is structured around a support set $\mathcal{S} = \{(x_i^{\mathcal{S}}, y_i^{\mathcal{S}})\}_{i=1}^{NK}$, adhering to the ($N$-way, $K$-shot) scheme, where $K$ labeled examples from $N$ distinct classes are chosen. The query set $Q = \{x_j^{Q}\}_{j=1}^{NQ}$, typically unlabeled, comprises $NQ$ samples (where $Q > K$) and serves to evaluate the model's adaptation to novel classes.

### 3.2 Unsupervised Pretraining Methods

The foundation of U-FSL is significantly influenced by the capabilities of unsupervised pretraining methods to discern intricate patterns within unlabeled datasets. This segment explores the effects of two principal unsupervised pretraining strategies: Contrastive Learning (CL) and Masked Image Modeling (MIM). Key exemplars for these methods—SimCLR [9] for CL and MAE [22] for MIM—were selected due to their prominence and efficacy. Both methodologies were implemented using the Vision Transformer Small (ViT-S) architecture on the MiniImageNet dataset. Our analytical focus is directed towards understanding how these pretraining approaches modify the model's capability to transition effectively to novel tasks. We postulate that the intrinsic nature of the pretraining method—contrastive, which underscores learning distinctive features that delineate classes, versus masked, which centers on reconstructing absent segments of the input—will manifest differing strengths within the context of few-shot learning.

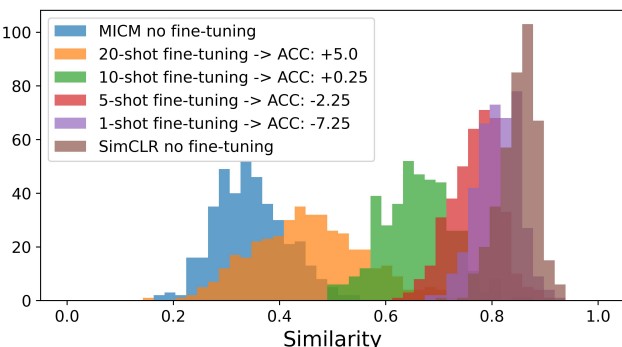

**Figure 3: Histograms depicting the distribution of similarity between features extracted by the model for novel and base classes. Our model (blue) extracts distinctive features for novel classes, contrasting with the SimCLR model from contrastive learning methods, which continues to focus on discriminative features of base classes for novel classes, resulting in similar features (brown). Fine-tuning with an adequate number of labeled samples is essential to address these issues in SimCLR and enhance classification accuracy.**

**Analysis of Contrastive Learning.** Recent research [12] has demonstrated that CL models tend to focus predominantly on the primary objects within images during pretraining. This concentration frequently targets small, distinct features that distinctly characterize the primary categories. Such specificity, while beneficial for initial categorization tasks, adversely affects the generalizability of the learned features to new, unseen contexts, as evidenced in Figure 1(b). We propose that this limitation arises from the inherent design of CL methodologies, which predispose the model to emphasize features that are salient in the training dataset but may be less relevant or even misleading in novel categories. To explore this proposition, we conducted empirical analyses comparing the feature representations of base category prototypes with those of novel category prototypes, both before and after applying SimCLR and its subsequent fine-tuning. The results, depicted in Figure 3, indicate that the features extracted for novel categories by SimCLR closely mirror those associated with the base categories. This similarity persists when models trained via CL are applied to novel categories, leading to a continued reliance on the same features identified during the training phase. Consequently, there is a notable deficiency in the model's focus on principal objects in new categories, which hampers its adaptability. Building on established protocols [9, 15], we fine-tuned the SimCLR-trained models on downstream few-shot classification tasks. Our findings, illustrated in Figure 3, show that fine-tuning with a minimal set of labeled examples (e.g., in one-shot learning scenarios) fails to adequately rectify these issues of transferability. In some instances, this approach may even degrade the model's performance on few-shot classification tasks. It becomes apparent that only with an ample number of labeled samples for fine-tuning do these challenges begin to mitigate, consequently improving accuracy in few-shot classification scenarios. This highlights the intrinsic difficulties of directly applying CL models to few-shot learning tasks, especially considering the heightened risk of overfitting when training data are scarce. Hence, we assert the following conclusion concerning the impact of contrastive learning:

***Conclusion and Discussion.*** Contrastive learning fundamentally predisposes models to prioritize features that are prominent in the training dataset, potentially at the expense of relevance and utility in novel categories.

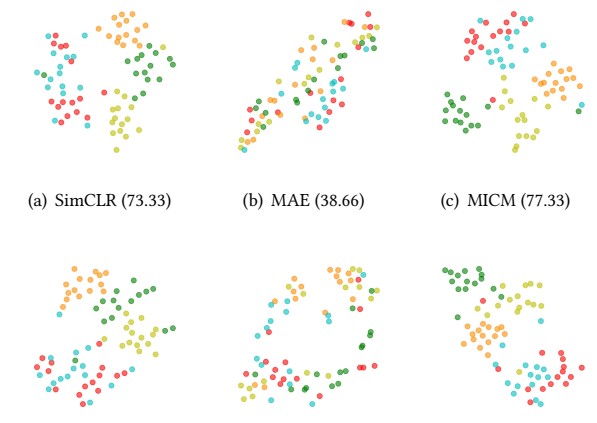

(a) SimCLR (73.33)    (b) MAE (38.66)    (c) MICM (77.33)

(d) SimCLR 5-shot (70.66)    (e) MAE 5-shot (41.33)    (f) MICM 5-shot (78.66)

**Figure 4: Visualization of t-SNE features for various unsupervised pre-trained models in novel categories. The first row: before fine-tuning. The second row: after fine-tuning with 5-shot. Contrary to SimCLR and MICM, the MAE method lacks discriminative features both prior to and following few-shot learning. The performance of each model in FSL is indicated in parentheses.**

**Analysis of Masked Image Modeling.** Recent advancements in MIM underscore a shift towards a more comprehensive feature extraction methodology, distinct from techniques that prioritize prominent image features. As detailed in [12], MIM methods like the MAE engage systematically with every image patch to reconstruct absent segments, fostering a broad spatial activation across the entire image. This approach is vividly illustrated in Figure 1(c), where feature maps from MAE reveal a widespread distribution of activation, suggesting a more holistic grasp of image features. Despite these strengths, the extensive focus on patch reconstruction in MIM can obscure class-specific feature delineation. Since MIM models are optimized for predicting missing image parts rather than distinguishing class features, they frequently lack the sharp, discriminative capabilities essential for class-specific recognition tasks. This deficiency is apparent in Figure 4, where the first row demonstrates that features extracted by MAE are markedly less discriminative than those derived from our proposed method or the contrastive learning approach, SimCLR. The adaptability of MIM techniques to FSL scenarios is also challenged when these models are fine-tuned with limited labeled data. The second row of Figure 4 indicates that, even after fine-tuning with a modest sample size, such as in a 5-shot scenario, the discriminability of the features shows minimal enhancement. This observation implies that while MIM effectively encodes a rich array of generic visual features, it struggles to capture the subtleties required for distinguishing novel classes in few-shot configurations.

***Conclusion and Discussion.*** While MIM techniques cultivate a broad and generalized understanding of image features, they encounter significant obstacles in acquiring discriminative features crucial for accurately categorizing novel classes in few-shot learning scenarios.

## 4 MASKED IMAGE CONTRASTIVE MODELING

The preceding comparative analysis in Section 3 elucidates that while CL prioritizes refining focus on distinct objects within datasets, enhancing representational efficacy for image classification, MIM extends its reach to a comprehensive understanding across all image patches, thus facilitating a broader scope of feature extraction. This delineation underscores a pivotal trade-off in unsupervised pretraining between generalization and discriminability. To bridge this gap, we introduce a novel approach, Masked Image Contrastive Modeling (MICM), which amalgamates the strengths of both methodologies to foster robust representation learning coupled with an effective FSL task.

### 4.1 Model Structure

Figure 2 illustrates the encoder-decoder architecture of MICM, ingeniously designed to predict masked patches based on visible ones within the encoded space, while concurrently ensuring similar tokens for identical images are decoded effectively. The image undergoes segmentation into visible patches $\mathbf{X}_v$ and masked patches $\mathbf{X}_m$. The encoder $\mathcal{F}$ processes $\mathbf{X}_v$ to generate latent representations $\mathbf{Z}_v$, while the decoder $\mathcal{G}$ aims to reconstruct $\mathbf{X}_m$ using these encoded representations along with a predefined class token $\mathbf{T}_c$.

**Encoder.** The encoder $\mathcal{F}$ transforms visible patches $\mathbf{X}_v$ into latent representations $\mathbf{Z}_v$. Utilizing the ViT as its backbone, the encoder begins by embedding the visible patches, projecting each patch linearly to create a set of embeddings. Positional embeddings $\mathbf{P}_v$ are added to maintain spatial context. These embeddings undergo processing through several transformer blocks, leveraging self-attention mechanisms to produce the latent representations $\mathbf{Z}_v$, which encapsulate the critical features of the visible patches.

**Decoder.** The decoder serves dual functions in MICM. Its primary role is to transform the latent representations of visible patches $\mathbf{Z}_v$ and, crucially, those of masked patches $\mathbf{Z}_m$ back into the reconstructed patches $\mathbf{Y}_m$. This transformation process entails a sequence of transformer blocks culminating in a linear layer that precisely regenerates the original masked patches. Secondly, the decoder also refines the input class token $\mathbf{Z}_{[\text{cls}]}$ into an enhanced representation $\hat{\mathbf{Z}}_{[\text{cls}]}$, pivotal for effective CL. Diverging from conventional approaches, MICM strategically delays the integration of the class token until the decoding phase, permitting the encoder to concentrate more thoroughly on capturing the nuances of visible patches. This structural delineation enhances the encoder's focus on extracting a diverse array of visual features, while the decoder, through feature reconstruction, fine-tunes the class token, synergistically balancing the model's objectives of maximizing discriminability and ensuring comprehensive feature extraction.

### 4.2 U-FSL with MICM

**Unsupervised Pretraining.** Given an input image $x$ uniformly sampled from the training set $D_{\text{base}}$, we apply random data augmentations to create two distinct views $x^1$ and $x^2$. These views are subsequently processed by the teacher and student networks,

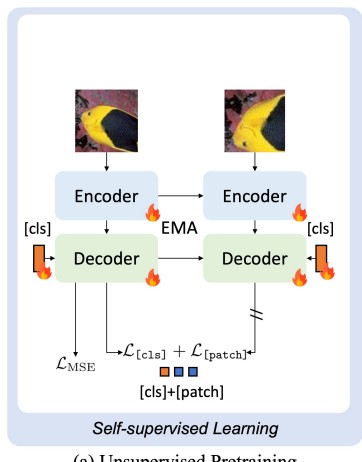
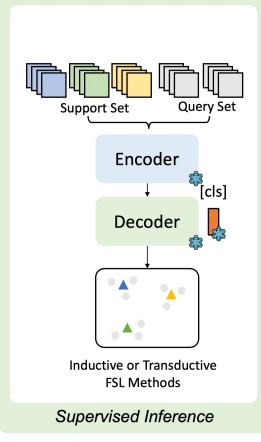

(a) Unsupervised Pretraining  (b) Few-Shot Learning

**Figure 5: (a) The *Unsupervised Pretraining* phase involves self-supervised learning on a large, unlabeled *base* dataset, crucial for developing initial representations. (b) The *Few-Shot Learning* phase could adapt to a variety of few-shot learning approaches, including both inductive and transductive methodologies.**

parameterized by $\theta^t$ and $\theta^s$, respectively. Notably, the teacher network is updated via an Exponentially Moving Average (EMA) of the student network, facilitating knowledge transfer by minimizing the cross-entropy loss between the output categorical distributions of their augmented token representations, as expressed in the following equation:

$$\mathcal{L}_{[\texttt{cls}]} = \mathcal{H}(\hat{\mathbf{Z}}^t_{[\texttt{cls}]}, \hat{\mathbf{Z}}^s_{[\texttt{cls}]}), \qquad (1)$$

where $\mathcal{H}(x, y) = -x \log y$, and $\hat{\mathbf{Z}}_{[\texttt{cls}]}$ denotes the output class token. For MIM, we implement self-distillation as proposed in [50]. A random mask sequence $m \in \{0, 1\}^M$ is applied over an image with $N_{[\texttt{patch}]}$ tokens $\boldsymbol{x} = \{\boldsymbol{x}_i\}_{i=1}^M$. The masked patches, denoted by $m_i = 1$, are replaced by a learnable token embedding $\mathbf{Z}_m$, resulting in a corrupted image $\widehat{\boldsymbol{x}}$. The student and teacher networks receive the corrupted and original uncorrupted images, respectively, to recover the masked tokens. This is quantified by minimizing the cross-entropy loss between their categorical distributions on masked patches:

$$\mathcal{L}_{[\texttt{patch}]} = \sum_{i=1}^M m_i \cdot \mathcal{H}(\hat{\mathbf{Z}}^t_{[\texttt{patch i}]}, \hat{\mathbf{Z}}^s_{[\texttt{patch i}]}). \qquad (2)$$

Moreover, we aim for the decoder-generated tokens to predict the RGB information of the image, incorporating an image reconstruction loss using Mean Squared Error (MSE) for the reconstruction targets $\bar{\mathbf{Y}}$:

$$\mathcal{L}_{\text{MSE}} = \sum (\mathbf{Y}_m, \bar{\mathbf{Y}}_m)^2. \qquad (3)$$

**Few-shot Learning.** During the few-shot learning phase, the MICM approach is adeptly configured to adapt to diverse few-shot learning strategies, encompassing both inductive and transductive methods. The MICM methodology enhances feature learning by emphasizing generality across base classes and discriminative power.

This dual capability significantly boosts the transferability of the learned features to few-shot tasks, thereby enabling superior adaptation to scenarios with limited labeled data. Further exploration of this adaptability is discussed in subsequent sections.

## 5 EXPERIMENTS

### 5.1 Datasets

Unsupervised few-shot recognition experiments are conducted on three benchmark datasets widely recognized in the field: MiniImageNet [41], TieredImageNet [35], and CIFAR-FS [5]. MiniImageNet, derived from the larger ILSVRC-12 dataset [36], consists of 100 categories, each represented by 600 images. It is divided into meta-training, meta-validation, and meta-testing segments, containing 64, 16, and 20 categories, respectively. TieredImageNet, also a subset of ILSVRC-12, includes 608 categories segmented into 351, 97, and 160 categories for training, validation, and testing. CIFAR-FS, a subset of CIFAR100 [28], follows a similar structure to MiniImageNet, with 60,000 images spread across 100 categories. These datasets provide a robust framework for evaluating few-shot learning algorithms. Additionally, cross-domain experiments use MiniImageNet as the pretraining (source) dataset and ISIC [16], EuroSAT [24], and CropDiseases [43] as inference (target) datasets, enhancing the generalizability assessment of the models.

### 5.2 Implementation Details

**Self-Supervised Learning:** The Vision Transformer (ViT) backbone and its associated projection head are pre-trained following the iBOT [50] framework, retaining most original hyper-parameters. The training employs a batch size of 640 and a learning rate of 0.0005 on a cosine decay schedule. The MiniImageNet and TieredImageNet datasets are pre-trained for 1200 epochs, while CIFAR-FS is pre-trained for 950 epochs. Additional training details are provided in the appendix.

**Few-shot Evaluation:** The pre-trained ViT backbone functions as the feature extractor. We utilize various few-shot learning (FSL) methods, including the prototypical networks approach [39], for evaluation. In each N-way K-shot task, class prototypes are calculated as the mean of the features from K support samples per class. Query images are then classified based on the highest cosine similarity to these prototypes. The feature set for evaluation combines the [cls] token with the weighted average [patch] token, using self-attention values from the last transformer layer. Test accuracies are reported over 2000 episodes, with each episode featuring 15 query shots per class, consistent with standard practices in the literature [10, 32, 33], and presented with 95% confidence intervals for all datasets.

### 5.3 Analysis of MICM

**Discriminative and Generalization Capabilities.** We investigate the critical balance between generalization and discriminability in unsupervised pretraining through our proposed MICM model. As depicted in Figure 6, MICM surpasses other unsupervised pretraining methods in capturing comprehensive object information from novel classes, exhibiting superior overall perception. This is further corroborated by the distance distribution between the prototypes of novel and base classes in Figure 3, where MICM's prototypes

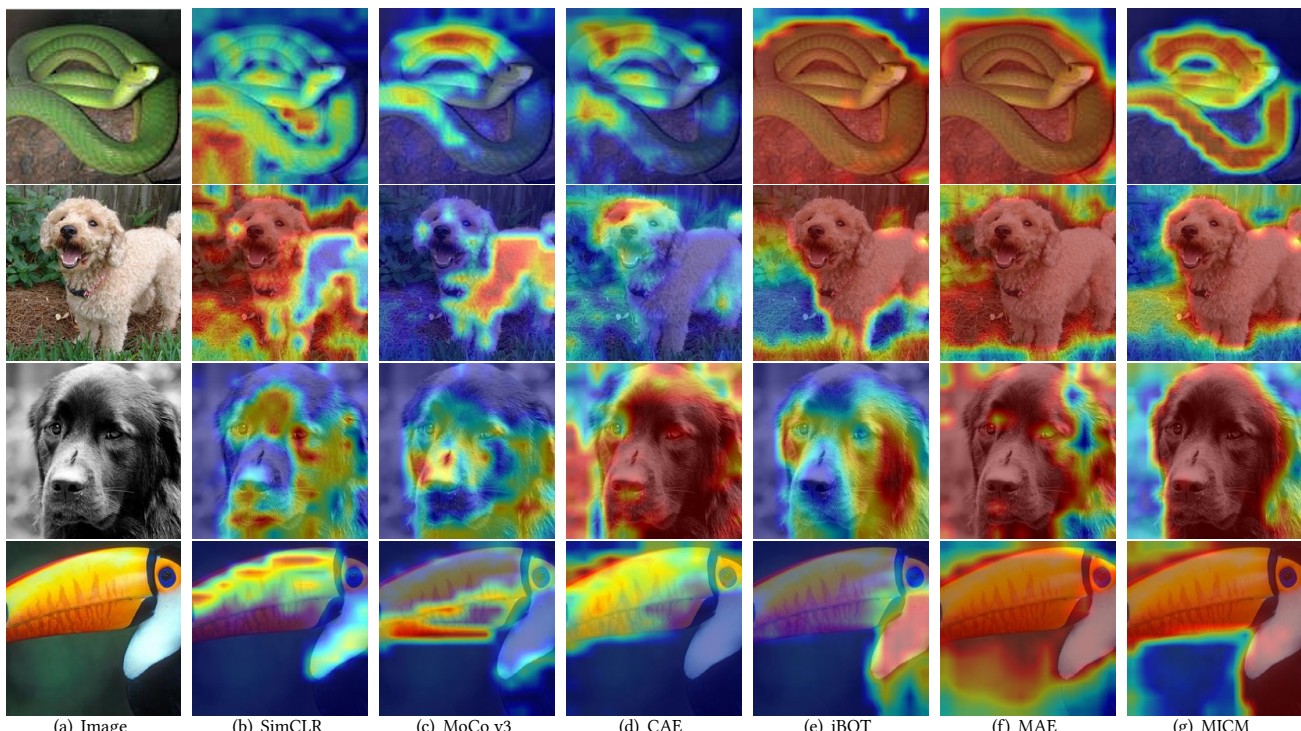

(a) Image    (b) SimCLR    (c) MoCo v3    (d) CAE    (e) iBOT    (f) MAE    (g) MICM

**Figure 6: Attention map visualization for different unsupervised pre-trained models. This figure presents a series of columns, each corresponding to the attention map output from a distinct model. From left to right, the columns are as follows: (a) the original image; (b) SimCLR [9]; (c) Moco v3 [15]; (d) CAE [12]; (e) iBOT [51]; (f) MAE [22]; and (g) MICM (ours).**

**Table 1: Accuracies (in % ± standard deviation) on miniImageNet, comparing our model with various unsupervised pretraining methods (all models use VIT-S as backbone). CTB denotes the strategy of inserting a classification *cls token* before the processing by the encoder.**

| Method | Setting | Inductive (ProtoNet) [39] | | Transductive (OpTA) [33] | |
|---|---|---|---|---|---|
| | | 5-way 1-shot | 5-way 5-shot | 5-way 1-shot | 5-way 5-shot |
| SimCLR [9] | CL | 54.30±0.62 | 75.03±0.35 | 65.83±0.64 | 78.09±0.40 |
| MoCo V3 [23] | CL | 56.06±0.43 | 76.78±0.33 | 71.45±0.64 | 82.04±0.35 |
| MAE [22] | MIM | 29.11±0.44 | 37.01±0.31 | 25.36±0.48 | 35.21±0.42 |
| CAE [12] | MIM | 57.33±0.46 | 79.25±0.33 | 70.34±0.67 | 81.36±0.39 |
| iBOT [50] | MIM | 60.93±0.21 | 80.38±0.16 | 74.58±0.66 | 83.95±0.34 |
| **MICM w/ CTB** | MIM+CL | **62.85±0.17** | **82.37±0.11** | 77.89±0.62 | 86.36±0.33 |
| **MICM** | MIM+CL | 60.78±0.19 | 81.39±0.14 | **78.40±0.61** | **86.90±0.33** |

**Table 2: Accuracies (in % ± std) on miniImageNet → CUB., comparing our model with various unsupervised pretraining methods (all models use VIT-S as backbone). CTB denotes the strategy of inserting a classification *cls token* before the processing by the encoder.**

| Method | Setting | Inductive (ProtoNet) [39] | | Transductive (OpTA) [33] | |
|---|---|---|---|---|---|
| | | 5-way 1-shot | 5-way 5-shot | 5-way 1-shot | 5-way 5-shot |
| SimCLR [9] | CL | 39.80±0.32 | 55.72±0.37 | 38.99±0.40 | 53.09±0.42 |
| MoCo V3 [23] | CL | 42.12±0.33 | 59.33±0.37 | 41.83±0.41 | 57.36±0.42 |
| MAE [22] | MIM | 30.13±0.25 | 37.94±0.31 | 25.37±0.25 | 31.89±0.32 |
| CAE [12] | MIM | 38.10±0.43 | 51.56±0.53 | 38.31±0.56 | 49.01±0.58 |
| iBOT [50] | MIM | 42.71±0.33 | 59.33±0.38 | 43.30±0.43 | 58.56±0.43 |
| **MICM w/ CTB** | MIM+CL | **45.06±0.34** | 62.83±0.37 | 46.85±0.45 | 62.75±0.42 |
| **MICM** | MIM+CL | 44.95±0.34 | **63.05±0.37** | **47.42±0.46** | **63.86±0.42** |

distinctly differ more from the base class, affirming its enhanced generalization for novel classes. Additionally, Figure 4 demonstrates MICM's superior discriminative capabilities compared to models like MAE, with a clearer distinction in feature distributions between MICM and SimCLR. Notably, MICM maintains leading performance in small-sample scenarios, both pre- and post-fine-tuning. The experimental validations highlight MICM's adept integration of the strengths of CL and MIM, achieving remarkable discriminability and generalization.

**Improving FSL.** Our exploration focuses on the synergy between MIM and CL, designed to overcome the limitations inherent to each

approach individually. The MICM model we introduce effectively integrates the strengths of these methodologies, emphasizing the extraction of relevant feature scales within images. This integration not only enhances category discrimination but also bolsters robustness in subsequent FSL tasks. The effectiveness of MICM is demonstrated through its superior performance in both inductive and transductive few-shot classification settings, detailed in Table 1.

**Enhancing Cross-Domain FSL.** In cross-domain scenarios, MICM also significantly excels, notably on the CUB dataset (Table 2). Unlike traditional CL models, which often overfit to base classes, MICM maintains generalization across varied domains without the need

**Table 3: Accuracies (in % ± standard deviation) on miniImageNet, comparing our model with various unsupervised pretraining methods, which are adapted to several FSL methods [11, 33, 39, 44, 46].**

| Pretrained Model | FSL method | 5-way 1-shot | 5-way 5-shot |
|---|---|---|---|
| MAE [22] | | 28.88 ± 0.43 | 37.19 ± 0.51 |
| SimCLR [9] | ProtoNet [39] | 54.42 ± 0.66 | 75.03 ± 0.35 |
| iBOT [50] | | 61.26 ± 0.66 | 80.64 ± 0.45 |
| **MICM** | | **61.37 ± 0.62** | **81.68 ± 0.43** |
| MAE [22] | | 28.50 ± 0.32 | 38.29 ± 0.50 |
| SimCLR [9] | Fine-tuning [11] | 54.47 ± 0.59 | 75.01 ± 0.36 |
| iBOT [50] | | 61.11 ± 0.59 | 80.91 ± 0.39 |
| **MICM** | | **61.41 ± 0.52** | **81.72 ± 0.32** |
| MAE [22] | | 30.30 ± 0.47 | 38.65 ± 0.50 |
| SimCLR [9] | SimpleShot [44] | 57.13 ± 0.64 | 74.88 ± 0.46 |
| iBOT [50] | | 61.98 ± 0.65 | 80.56 ± 0.45 |
| **MICM** | | **62.53 ± 0.63** | **81.79 ± 0.43** |
| MAE [22] | | 37.06 ± 0.47 | 52.95 ± 0.51 |
| SimCLR [9] | DC [46] | 60.86 ± 0.58 | 75.79 ± 0.39 |
| iBOT [50] | | 65.84 ± 0.67 | 83.77 ± 0.43 |
| **MICM** | | **67.19 ± 0.65** | **85.12 ± 0.41** |
| MAE [22] | | 25.36 ± 0.48 | 35.21 ± 0.42 |
| SimCLR [9] | OpTA [33] | 65.83 ± 0.64 | 78.09 ± 0.40 |
| iBOT [50] | | 74.58 ± 0.66 | 83.95 ± 0.34 |
| **MICM** | | **78.40 ± 0.61** | **86.90 ± 0.33** |

for fine-tuning, as illustrated in Figures 3 and 4. This capability underscores MICM's effectiveness in capturing discernible features within an optimal range, thus boosting its adaptability and classification performance in few-shot learning across different domains.

**Broad Adaptation to FSL Methods.** As a versatile pre-training model, MICM adapts seamlessly across a spectrum of FSL strategies. Comprehensive evaluations show that MICM invariably boosts performance, with notable improvements such as a nearly 4% enhancement over the iBOT model when utilizing the transductive method OpTA. These results affirm the robust generalization ability of MICM across a range of FSL approaches.

*cls token* **Variation.** In exploring variations, we introduced a *cls token* as an input to the Encoder, with performance outcomes detailed in Tables 1 and 2. Although this variant achieves commendable results in the inductive setting, it does not outperform the configuration where the *cls token* is input into the Decoder, especially in transductive scenarios. This suggests that introducing the *cls token* early in the encoder may impede the encoder's ability to learn comprehensive visual features effectively. Conversely, positioning the *cls token* in the decoder helps alleviate potential negative impacts by CL on learning holistic visual features.

## 5.4 Comparison with SOTA Method

To assess the efficacy of MICM in FSL scenarios, particularly under the U-FSL framework, we developed and evaluated a novel methodology that combines unsupervised pre-training with pseudo-label training techniques. We integrated pseudo-label learning [33] with the transductive OpTA FSL method [33], forming a hybrid approach designed to leverage the combined strengths of these methods to boost performance in scenarios with scarce labeled data. Our

method's performance was benchmarked against SOTA models across various datasets, with detailed methodological descriptions provided in the Appendix.

**In-Domain Setting.** Our model was evaluated against a broad range of baselines including established SSL baselines [7, 9, 14, 19, 20, 48], prominent U-FSL methods [8, 10, 25, 27, 32, 37], leading supervised FSL approaches [3, 30], and a recent transductive U-FSL model [33]. Our model demonstrates superior performance, outperforming both inductive and transductive U-FSL methods as evidenced in Tables 6 and 4, showing a notable accuracy improvement on the CIFAR-FS dataset.

Our implementation utilizes the ViT architecture, which contrasts with the commonly used ResNet in U-FSL studies. To facilitate comprehensive evaluation, we compared results from models using both ResNet18 and ResNet50 architectures, and additionally, we benchmarked against a ViT-S model trained using the MIM method for transductive classification (MIM+OpTA), providing a baseline for ViT-based transductive U-FSL models.

**Table 4: Accuracies (in % ± std) for CIFAR-FS dataset.**

| Method | 5-way 1-shot | 5-way 5-shot |
|---|---|---|
| SimCLR [9] | 54.56±0.19 | 71.19±0.18 |
| MoCo v2 [13] | 52.73±0.20 | 67.81±0.19 |
| LF2CS [31] | 55.04±0.72 | 70.62±0.57 |
| HMS [47] | 54.65±0.20 | 73.70±0.18 |
| BECLR [33] | 70.39±0.62 | 81.56±0.39 |
| **MICM** | **79.20±0.61** | **86.35±0.39** |

Regarding CIFAR-FS performance comparisons (Table 4), sourced from [33], we note that the BECLR model reported results using ResNet18. Hence, our model's reported performance is achieved with a scaled-down version of ViT-S, comprising 6 layers (4 encoder layers and 2 decoder layers) as opposed to the full 12 layers in standard ViT-S.

**Table 5: 5-way 5-shots accuracies (in % ± std) on miniImageNet → Cross-Domain Few-Shot Learning.**

| Method | ChestX | ISIC | EuroSAT | CropDiseases | Mean |
|---|---|---|---|---|---|
| SwAV [7] | 25.70±0.28 | 40.69±0.34 | 84.82±0.24 | 88.64±0.26 | 60.12 |
| NNCLR [19] | 25.74±0.41 | 38.85±0.56 | 83.45±0.53 | 90.76±0.57 | 59.70 |
| SAMPTransfer [37] | 26.27±0.44 | 47.60±0.59 | 85.55±0.60 | 91.74±0.55 | 62.79 |
| PsCo [27] | 24.78±0.23 | 44.00±0.30 | 81.08±0.35 | 88.24±0.31 | 59.52 |
| UniSiam + dist [32] | 28.18±0.45 | 45.65±0.58 | 86.53±0.47 | 92.05±0.50 | 63.10 |
| ConFeSS [17] | 27.09 | **48.85** | 84.65 | 88.88 | 62.36 |
| BECLR [33] | **28.46±0.23** | 44.48±0.31 | 88.55±0.23 | 93.65±0.25 | 63.78 |
| MICM | 27.11±0.36 | 46.85±0.52 | **90.08±0.36** | **94.61±0.27** | **64.66** |

**Cross-Domain Setting.** Following established methodologies [21, 33], we pretrained on the miniImageNet dataset and evaluated our approach in cross-domain few-shot learning settings. The results, detailed in Table 5, demonstrate that MICM sets new SOTA benchmarks on the EuroSAT and Crop Diseases datasets, while maintaining competitive performance on the ISIC dataset. MICM's adaptive training mechanism enables superior performance over BECLR in cross-domain settings, highlighting its robustness and adaptability across diverse datasets.

**Table 6: Accuracies (in % ± standard deviation) on miniImageNet and tieredImageNet, comparing our model with various baselines categorized into Inductive (Ind.) and Transductive (Transd.) approaches. Performance is delineated by backbone architectures, namely Residual Networks (RN) and Vision Transformers (ViT), with the number of parameters (Param) for each model included for an extensive comparison.**

| Method | Backbone | Param | Setting | miniImageNet | | tieredImageNet | |
|---|---|---|---|---|---|---|---|
| | | | | 5-way 1-shot | 5-way 5-shot | 5-way 1-shot | 5-way 5-shot |
| SwAV [7] | RN18 (×1) | 11.2M | Ind. | 59.84 ± 0.52 | 78.23 ± 0.26 | 65.26 ± 0.53 | 81.73 ± 0.24 |
| NNCLR [19] | RN18 (×2) | 22.4M | Ind. | 63.33 ± 0.53 | 80.75 ± 0.25 | 65.46 ± 0.55 | 81.40 ± 0.27 |
| CPNWCP [42] | RN18 (×1) | 11.2M | Ind. | 53.14 ± 0.62 | 67.36 ± 0.5 | 45.46 ± 0.19 | 62.96 ± 0.19 |
| HMS [47] | RN18 (×1) | 11.2M | Ind. | 58.20 ± 0.23 | 75.77 ± 0.16 | 58.42 ± 0.25 | 75.85 ± 0.18 |
| SAMPTransfer [37] | RN18 (×1) | 11.2M | Ind. | 45.75 ± 0.77 | 68.33 ± 0.66 | 42.32 ± 0.75 | 53.45 ± 0.76 |
| PsCo [27] | RN18 (×1) | 11.2M | Ind. | 47.24 ± 0.76 | 65.48 ± 0.68 | 54.33 ± 0.54 | 69.73 ± 0.49 |
| PDA-Net [10] | RN50 (×1) | 23.5M | Ind. | 63.84 ± 0.91 | 83.11 ± 0.56 | 69.01 ± 0.93 | 84.20 ± 0.69 |
| UniSiam + dist [32] | RN50 (×1) | 23.5M | Ind. | 65.33 ± 0.36 | 83.22 ± 0.24 | 69.60 ± 0.38 | 86.51 ± 0.26 |
| Meta-DM + UniSiam + dist [25] | RN50 (×1) | 23.5M | Ind. | 66.68 ± 0.36 | 85.29 ± 0.23 | 69.61 ± 0.38 | 86.53 ± 0.26 |
| CPNWCP + OpTA [42] | RN18 (×1) | 11.2M | Transd. | 60.45 ± 0.81 | 75.84 ± 0.56 | 55.05 ± 0.31 | 72.91 ± 0.26 |
| HMS + OpTA [47] | RN18 (×1) | 11.2M | Transd. | 69.85 ± 0.42 | 80.77 ± 0.35 | 71.75 ± 0.43 | 81.32 ± 0.34 |
| PsCo + OpTA [27] | RN18 (×1) | 11.2M | Transd. | 52.89 ± 0.71 | 67.42 ± 0.54 | 57.46 ± 0.59 | 70.70 ± 0.45 |
| UniSiam + OpTA [32] | RN18 (×1) | 11.2M | Transd. | 72.54 ± 0.61 | 82.46 ± 0.32 | 73.37 ± 0.64 | 73.37 ± 0.64 |
| BECLR [33] | RN18 (×2) | 22.4M | Transd. | 75.74 ± 0.62 | 84.93 ± 0.33 | 76.44 ± 0.66 | 84.85 ± 0.37 |
| BECLR [33] | RN50 (×2) | 47M | Transd. | 80.57 ± 0.57 | 87.82 ± 0.29 | 81.69 ± 0.61 | 87.82 ± 0.32 |
| **MICM** | **VIT-S (×2)** | **42M** | **Transd.** | **81.05 ± 0.58** | **87.95 ± 0.34** | **83.30 ± 0.61** | **89.61 ± 0.35** |

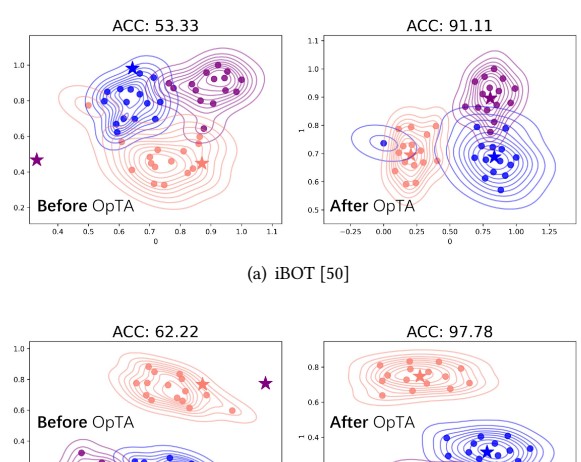

(a) iBOT [50]

(b) MICM

**Figure 7: Feature distribution maps comparing various methods before and after applying OpTA.**

**Feature Distribution Analysis.** To deepen our understanding of the MICM mechanism, we employed iBOT [50] as a baseline for comparative analysis. A critical observation, illustrated in Figure 7, is that MICM significantly enhances the compactness and cohesion of feature distributions within the same category. Compared to the baseline, where feature clusters are dispersed and misaligned (as shown in Figure 7(a)), our model demonstrates a notably tighter

clustering. This improvement is especially evident in the alignment of support samples with the corresponding query samples within each category. The application of the OpTA method notably rectifies sample bias, further refining feature distribution, and alignment. This adjustment, combined with the advanced feature representation capabilities of our MICM model, yields a substantial enhancement in performance relative to the baseline. The precise clustering of category-specific features and the effective mitigation of sample bias by OpTA underline the robustness and effectiveness of our model in generating highly discriminative feature representations, which is pivotal for few-shot learning applications.

## 6 CONCLUSION

In this paper, we have delineated the limitations of Masked Image Modeling (MIM) and Contrastive Learning (CL) in terms of their discriminative and generalization capabilities, which have contributed to their underperformance in Unsupervised Few-Shot Learning (U-FSL) contexts. To tackle these challenges, we introduced Masked Image Contrastive Modeling (MICM), a novel approach that effectively integrates the strengths of MIM and CL. Our results demonstrate that MICM adeptly balances discriminative power with generalizability, particularly in few-shot learning scenarios characterized by limited sample sizes. MICM's flexibility in adapting to various few-shot learning strategies highlights its potential as a versatile and powerful tool for unsupervised pretraining within the U-FSL framework. Extensive quantitative and qualitative evaluations show MICM's clear superiority over existing methods, confirming its ability to enhance feature discrimination, robustness, and adaptability across diverse few-shot learning tasks.

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
