# OpenReview forum: "MICM: Rethinking Unsupervised Pretraining for Enhanced Few-shot Learning"
_acmmm.org/ACMMM/2024/Conference — MM2024 Oral_

### Official Review · Reviewer_yuKq · 2024-05-20

**Rating:** 5
**Confidence:** 4

**Summary:**

This paper propose Masked Image Contrastive Modeling (MICM), a novel approach that effectively integrates the strengths of MIM and CL. Quantitative and  qualitative results demonstrate that MICM adeptly balances discriminative power with generalizability, particularly in few-shot learning scenarios characterized by limited sample sizes.

**Strengths:**

+ The paper is well-written and easy to understand
+ The figures are well-designed and clearly express the ideas.
+ Both qualitative and quantitative results are provided.
+ The proposed method is orthogonal existing transductive and inductive FSL methods.

**Limitations:**

+  The computational complexity and well as the limitations of the proposed method should be further discussed.
+ Leveraging the idea of contrastive learning to establish more valuable representations has been explored in FSL [1][2]. Therefore, the differences w.r.t. those methods should be further discussed.

To the best of my knowledge, it is hard to find other major limitations in this paper.

[1] Free-lunch for Cross-domain Few-shot learning: Style-aware Episodic Training with Robust Contrastive Learning, ACM MM'22 \
[2] DETA: Denoised Task Adaptation for Few-Shot Learning, ICCV'23

**Suitability:**

2

---

### Official Review · Reviewer_cMKD · 2024-05-23

**Rating:** 5
**Confidence:** 2

**Summary:**

This paper investigates the limitations of masked image modeling (MIM) and contrastive learning (CL) in terms of discriminative and generalization abilities in a few-shot scenario and proposes a novel unsupervised pretraining framework named MICM, which comprises the advantages of both MIM and CL. The approach leads to improvements in several FSL datasets.

**Strengths:**

1. The paper identifies a crucial shortcoming of MIM and CL to few-shot learning; the features they extract need to be more balanced in discriminative and generalization. The analysis is intuitive and convincing.

2. The proposed method mitigates the above problem, providing a novel solution to capture better features for few-shot tasks.

**Limitations:**

1. It is necessary to explain the method of generating attention maps in Figures 1 & 6. Are they generating from techniques like GradCAM? Also, which attention layers of competing methods are being visualized?

2. Conducting experiments similar to those in Figure 3 for MAE would be beneficial. Will MAE without fine-tuning also provide features with high similarity to base categories due to the generaliztion ability?

3. Sec 4.2 illustrates the workflow using a positive image pair x1 and x2 as input. How can MICM develop discriminative ability (like SimCLR) without the use of negative image pairs?

**Suitability:**

3

---

### Official Review · Reviewer_27Dw · 2024-05-28

**Rating:** 5
**Confidence:** 3

**Summary:**

The paper presents a novel approach called Masked Image Contrastive Modeling (MICM) which integrates aspects of both Contrastive Learning and Masked Image Modeling to address the limitations in unsupervised Few-Shot Learning. The primary goal is to enhance the discriminative power and generalizability of models in few-shot scenarios by rethinking unsupervised pretraining. MICM uses a unique encoder-decoder architecture where the encoder focuses on visible patches of images to generate latent representations, and the decoder reconstructs masked patches while ensuring similarity in the representation space. This method significantly improves generalization and discrimination capabilities in downstream few-shot learning tasks, as demonstrated through extensive experimental analyses on various datasets.

**Strengths:**

1. The paper is generally well-written, with clear motivation and discussion.
2. The proposed approach is simple and effective.
3. The analysis in Section 3.2 is interesting.

**Limitations:**

1. The proposed approach can serve as a general unsupervised pre-training objective. It would be more interesting to see its performance on additional downstream tasks (other than unsupervised FSL).
2. It would be helpful to include a comparison with [1], which also aims to combine contrastive learning and MIM.

[1] Layer Grafted Pre-training: Bridging Contrastive Learning And Masked Image Modeling For Label-Efficient Representations. ICLR 2023.

**Suitability:**

2

---

### Official Review · Reviewer_GfLx · 2024-05-31

**Rating:** 3
**Confidence:** 3

**Summary:**

This paper explores the difference between human and machine learning, particularly in learning from few labeled samples. Unsupervised Few-Shot Learning (U-FSL) aims to reduce reliance on annotated datasets. The study evaluates Masked Image Modeling (MIM) and Contrastive Learning (CL) for few-shot learning, finding limitations in their discriminative and generalization abilities. To address these issues, the authors introduce Masked Image Contrastive Modeling (MICM), combining the strengths of CL and MIM. MICM improves performance in few-shot learning by enhancing generalization and discrimination. Experimental results confirm the effectiveness of MICM in these tasks.

**Strengths:**

This paper identifies the limitations of Masked Image Modeling (MIM) and Contrastive Learning (CL) in terms of their discriminative and generalization abilities, which lead to their underperformance in Unsupervised Few-Shot Learning (U-FSL). To address these shortcomings, the authors propose a novel approach called Masked Image Contrastive Modeling (MICM). MICM combines the object learning strengths of CL with the generalized feature learning capabilities of MIM, aiming to enhance performance in U-FSL tasks.

**Limitations:**

1. MICM seems to be a general self-supervised learning method. Why is it special for Few-shot Learning?
2. MICM is not novel. A lot of works[1] combine MAE and contrastive learning.

[1] Huang, Z., Jin, X., Lu, C., Hou, Q., Cheng, M. M., Fu, D., ... & Feng, J. (2023). Contrastive masked autoencoders are stronger vision learners. IEEE Transactions on Pattern Analysis and Machine Intelligence.

**Suitability:**

2

---

### Meta-Review · Area_Chair_rTxo · 2024-07-02

**Recommendation:** Accept (Oral)
**Confidence:** 4

**Metareview:**

The paper presents a novel method that integrates aspects of both contrastive learning and masked image modeling to address the limitations of unsupervised few-shot learning.

This paper received consistent positive ratings, and all reviewers confirmed that their concerns had been addressed during the rebuttal period. After reading the paper, reviews, and rebuttal document, I decided to recommend this paper for an oral presentation.